# Practical Bayesian Optimization for Model Fitting with Bayesian Adaptive Direct Search

**Luigi Acerbi**[*]
Center for Neural Science
New York University
`luigi.acerbi@nyu.edu`

**Wei Ji Ma**
Center for Neural Science & Dept. of Psychology
New York University
`weijima@nyu.edu`

## Abstract

Computational models in fields such as computational neuroscience are often evaluated via stochastic simulation or numerical approximation. Fitting these models implies a difficult optimization problem over complex, possibly noisy parameter landscapes. Bayesian optimization (BO) has been successfully applied to solving expensive black-box problems in engineering and machine learning. Here we explore whether BO can be applied as a general tool for model fitting. First, we present a novel hybrid BO algorithm, Bayesian adaptive direct search (BADS), that achieves competitive performance with an affordable computational overhead for the running time of typical models. We then perform an extensive benchmark of BADS vs. many common and state-of-the-art nonconvex, derivative-free optimizers, on a set of model-fitting problems with real data and models from six studies in behavioral, cognitive, and computational neuroscience. With default settings, BADS consistently finds comparable or better solutions than other methods, including 'vanilla' BO, showing great promise for advanced BO techniques, and BADS in particular, as a general model-fitting tool.

## 1  Introduction

Many complex, nonlinear computational models in fields such as behaviorial, cognitive, and computational neuroscience cannot be evaluated analytically, but require moderately expensive numerical approximations or simulations. In these cases, finding the maximum-likelihood (ML) solution – for parameter estimation, or model selection – requires the costly exploration of a rough or noisy nonconvex landscape, in which gradients are often unavailable to guide the search.

Here we consider the problem of finding the (global) optimum $x^* = \operatorname{argmin}_{x \in \mathcal{X}} \mathbb{E}\left[f(x)\right]$ of a possibly noisy *objective* $f$ over a (bounded) domain $\mathcal{X} \subseteq \mathbb{R}^D$, where the function $f$ can be intended as the (negative) log likelihood of a parameter vector $x$ for a given dataset and model, but is generally a *black box*. With many derivative-free optimization algorithms available to the researcher [1], it is unclear which one should be chosen. Crucially, an inadequate optimizer can hinder progress, limit the complexity of the models that can be fit, and even cast doubt on the reliability of one's findings.

*Bayesian optimization* (BO) is a state-of-the-art machine learning framework for optimizing expensive and possibly noisy black-box functions [2, 3, 4]. This makes it an ideal candidate for solving difficult model-fitting problems. Yet there are several obstacles to a widespread usage of BO as a general tool for model fitting. First, traditional BO methods target *very* costly problems, such as hyperparameter tuning [5], whereas evaluating a typical behavioral model might only have a moderate computational cost (e.g., 0.1-10 s per evaluation). This implies major differences in what is considered an acceptable algorithmic overhead, and in the maximum number of allowed function evaluations (e.g., hundreds vs.

---

[*]Current address: Département des neurosciences fondamentales, Université de Genève, CMU, 1 rue Michel-Servet, 1206 Genève, Switzerland. E-mail: `luigi.acerbi@gmail.com`.

thousands). Second, it is unclear how BO methods would fare in this regime against commonly used and state-of-the-art, non-Bayesian optimizers. Finally, BO might be perceived by non-practitioners as an advanced tool that requires specific technical knowledge to be implemented or tuned.

We address these issues by developing a novel hybrid BO algorithm, Bayesian Adaptive Direct Search (BADS), that achieves competitive performance at a small computational cost. We tested BADS, together with a wide array of commonly used optimizers, on a novel benchmark set of model-fitting problems with real data and models drawn from studies in cognitive, behaviorial and computational neuroscience. Finally, we make BADS available as a free MATLAB package with the same user interface as existing optimizers and that can be used out-of-the-box with no tuning.[1]

BADS is a *hybrid* BO method in that it combines the mesh adaptive direct search (MADS) framework [6] (Section 2.1) with a BO search performed via a local Gaussian process (GP) surrogate (Section 2.2), implemented via a number of heuristics for efficiency (Section 3). BADS proves to be highly competitive on both artificial functions and real-world model-fitting problems (Section 4), showing promise as a general tool for model fitting in computational neuroscience and related fields.

**Related work**    There is a large literature about (Bayesian) optimization of expensive, possibly stochastic, computer simulations, mostly used in machine learning [3, 4, 5] or engineering (known as *kriging-based* optimization) [7, 8, 9]. Recent work has combined MADS with treed GP models for constrained optimization (TGP-MADS [9]). Crucially, these methods have large overheads and may require problem-specific tuning, making them impractical as a generic tool for model fitting. Cheaper but less precise surrogate models than GPs have been proposed, such as random forests [10], Parzen estimators [11], and dynamic trees [12]. In this paper, we focus on BO based on traditional GP surrogates, leaving the analysis of alternative models for future work (see Conclusions).

## 2    Optimization frameworks

### 2.1    Mesh adaptive direct search (MADS)

The MADS algorithm is a directional direct search framework for nonlinear optimization [6, 13]. Briefly, MADS seeks to improve the current solution by testing points in the neighborhood of the current point (the *incumbent*), by moving one step in each direction on an iteration-dependent mesh. In addition, the MADS framework can incorporate in the optimization any arbitrary search strategy which proposes additional test points that lie on the mesh.

MADS defines the current mesh at the $k$-th iteration as $M_k = \bigcup_{\boldsymbol{x} \in S_k} \left\{ \boldsymbol{x} + \Delta_k^{\text{mesh}} \mathbf{D} \boldsymbol{z} : \boldsymbol{z} \in \mathbb{N}^D \right\}$, where $S_k \subset \mathbb{R}^n$ is the set of all points evaluated since the start of the iteration, $\Delta_k^{\text{mesh}} \in \mathbb{R}_+$ is the *mesh size*, and $\mathbf{D}$ is a fixed matrix in $\mathbb{R}^{D \times n_{\mathbf{D}}}$ whose $n_{\mathbf{D}}$ columns represent viable search directions. We choose $\mathbf{D} = [\mathbf{I}_D, -\mathbf{I}_D]$, where $\mathbf{I}_D$ is the identity matrix in dimension $D$.

Each iteration of MADS comprises of two stages, a SEARCH stage and an optional POLL stage. The SEARCH stage evaluates a finite number of points proposed by a provided search strategy, with the only restriction that the tested points lie on the current mesh. The search strategy is intended to inject problem-specific information in the optimization. In BADS, we exploit the freedom of SEARCH to perform Bayesian optimization in the neighborhood of the incumbent (see Section 2.2 and 3.3). The POLL stage is performed if the SEARCH fails in finding a point with an improved objective value. POLL constructs a *poll set* of candidate points, $P_k$, defined as $P_k = \left\{ \boldsymbol{x}_k + \Delta_k^{\text{mesh}} \boldsymbol{v} : \boldsymbol{v} \in \mathbf{D}_k \right\}$, where $\boldsymbol{x}_k$ is the incumbent and $\mathbf{D}_k$ is the set of *polling directions* constructed by taking discrete linear combinations of the set of directions $\mathbf{D}$. The *poll size* parameter $\Delta_k^{\text{poll}} \geq \Delta_k^{\text{mesh}}$ defines the maximum length of poll displacement vectors $\Delta_k^{\text{mesh}} \boldsymbol{v}$, for $\boldsymbol{v} \in \mathbf{D}_k$ (typically, $\Delta_k^{\text{poll}} \approx \Delta_k^{\text{mesh}} ||\boldsymbol{v}||$). Points in the poll set can be evaluated in any order, and the POLL is opportunistic in that it can be stopped as soon as a better solution is found. The POLL stage ensures theoretical convergence to a local stationary point according to Clarke calculus for nonsmooth functions [6, 14].

If either SEARCH or POLL are a *success*, finding a mesh point with an improved objective value, the incumbent is updated and the mesh size remains the same or is multiplied by a factor $\tau > 1$. If neither SEARCH or POLL are successful, the incumbent does not move and the mesh size is divided by $\tau$. The algorithm proceeds until a stopping criterion is met (e.g., maximum budget of function evaluations).

## 2.2 Bayesian optimization

The typical form of Bayesian optimization (BO) [2] builds a Gaussian process (GP) approximation of the objective $f$, which is used as a relatively inexpensive surrogate to guide the search towards regions that are promising (low GP mean) and/or unknown (high GP uncertainty), according to a rule, the acquisition function, that formalizes the exploitation-exploration trade-off.

**Gaussian processes** GPs are a flexible class of models for specifying prior distributions over unknown functions $f : \mathcal{X} \subseteq \mathbb{R}^D \to \mathbb{R}$ [15]. GPs are specified by a mean function $m : \mathcal{X} \to \mathbb{R}$ and a positive definite covariance, or kernel function $k : \mathcal{X} \times \mathcal{X} \to \mathbb{R}$. Given any finite collection of $n$ points $\mathbf{X} = \left\{ \boldsymbol{x}^{(i)} \in \mathcal{X} \right\}_{i=1}^{n}$, the value of $f$ at these points is assumed to be jointly Gaussian with mean $(m(\boldsymbol{x}^{(1)}), \ldots, m(\boldsymbol{x}^{(n)}))^\top$ and covariance matrix $\mathbf{K}$, where $\mathbf{K}_{ij} = k(\boldsymbol{x}^{(i)}, \boldsymbol{x}^{(j)})$ for $1 \leq i, j \leq n$. We assume i.i.d. Gaussian observation noise such that $f$ evaluated at $\boldsymbol{x}^{(i)}$ returns $y^{(i)} \sim \mathcal{N}\left(f(\boldsymbol{x}^{(i)}), \sigma^2\right)$, and $\boldsymbol{y} = (y^{(1)}, \ldots, y^{(n)})^\top$ is the vector of observed values. For a deterministic $f$, we still assume a small $\sigma > 0$ to improve numerical stability of the GP [16]. Conveniently, observation of such (noisy) function values will produce a GP posterior whose latent marginal conditional mean $\mu(\boldsymbol{x}; \{\mathbf{X}, \boldsymbol{y}\}, \boldsymbol{\theta})$ and variance $s^2(\boldsymbol{x}; \{\mathbf{X}, \boldsymbol{y}\}, \boldsymbol{\theta})$ at a given point are available in closed form (see Supplementary Material), where $\boldsymbol{\theta}$ is a hyperparameter vector for the mean, covariance, and likelihood. In the following, we omit the dependency of $\mu$ and $s^2$ from the data and GP parameters to reduce clutter.

**Covariance functions** Our main choice of stationary (translationally-invariant) covariance function is the automatic relevance determination (ARD) *rational quadratic* (RQ) kernel,

$$k_{\text{RQ}}\left(\boldsymbol{x}, \boldsymbol{x}'\right) = \sigma_f^2 \left[1 + \frac{1}{2\alpha} r^2(\boldsymbol{x}, \boldsymbol{x}')\right]^{-\alpha}, \qquad \text{with} \quad r^2(\boldsymbol{x}, \boldsymbol{x}') = \sum_{d=1}^{D} \frac{1}{\ell_d^2} \left(x_d - x_d'\right)^2, \quad (1)$$

where $\sigma_f^2$ is the signal variance, $\ell_1, \ldots, \ell_D$ are the kernel length scales along each coordinate direction, and $\alpha > 0$ is the shape parameter. More common choices for Bayesian optimization include the *squared exponential* (SE) kernel [9] or the twice-differentiable ARD *Matérn 5/2* ($\text{M}_{5/2}$) kernel [5], but we found the RQ kernel to work best in combination with our method (see Section 4.2). We also consider *composite periodic kernels* for circular or periodic variables (see Supplementary Material).

**Acquisition function** For a given GP approximation of $f$, the *acquisition function*, $a : \mathcal{X} \to \mathbb{R}$, determines which point in $\mathcal{X}$ should be evaluated next via a proxy optimization $\boldsymbol{x}_{\text{next}} = \text{argmin}_{\boldsymbol{x}} a(\boldsymbol{x})$. We consider here the *GP lower confidence bound* (LCB) metric [17],

$$a_{\text{LCB}}\left(\boldsymbol{x}; \{\mathbf{X}, \boldsymbol{y}\}, \boldsymbol{\theta}\right) = \mu\left(\boldsymbol{x}\right) - \sqrt{\nu \beta_t s^2\left(\boldsymbol{x}\right)}, \qquad \beta_t = 2 \ln\left(D t^2 \pi^2 / (6\delta)\right) \qquad (2)$$

where $\nu > 0$ is a tunable parameter, $t$ is the number of function evaluations so far, $\delta > 0$ is a probabilistic tolerance, and $\beta_t$ is a learning rate chosen to minimize cumulative regret under certain assumptions. For BADS we use the recommended values $\nu = 0.2$ and $\delta = 0.1$ [17]. Another popular choice is the (negative) *expected improvement* (EI) over the current best function value [18], and an historical, less used metric is the (negative) *probability of improvement* (PI) [19].

## 3 Bayesian adaptive direct search (BADS)

We describe here the main steps of BADS (Algorithm 1). Briefly, BADS alternates between a series of fast, local BO steps (the SEARCH stage of MADS) and a systematic, slower exploration of the mesh grid (POLL stage). The two stages complement each other, in that the SEARCH can explore the space very effectively, provided an adequate surrogate model. When the SEARCH repeatedly fails, meaning that the GP model is not helping the optimization (e.g., due to a misspecified model, or excess uncertainty), BADS switches to POLL. The POLL stage performs a fail-safe, model-free optimization, during which BADS gathers information about the local shape of the objective function, so as to build a better surrogate for the next SEARCH. This alternation makes BADS able to deal effectively and robustly with a variety of problems. See Supplementary Material for a full description.

### 3.1 Initial setup

**Problem specification** The algorithm is initialized by providing a starting point $\boldsymbol{x}_0$, vectors of *hard* lower/upper bounds LB, UB, and optional vectors of *plausible* lower/upper bounds PLB, PUB, with the

---

**Algorithm 1** Bayesian Adaptive Direct Search

---

**Input:** objective function $f$, starting point $\boldsymbol{x_0}$, hard bounds LB, UB, (*optional*: plausible bounds PLB, PUB, barrier function $c$, additional `options`)

1: **Initialization:** $\Delta_0^{\text{mesh}} \leftarrow 2^{-10}$, $\Delta_0^{\text{poll}} \leftarrow 1$, $k \leftarrow 0$, evaluate $f$ on initial design     ▷ Section 3.1
2: **repeat**
3:     (update GP approximation at any step; refit hyperparameters if necessary)     ▷ Section 3.2
4:     **for** $1 \ldots n_{\text{search}}$ **do**                                 ▷ SEARCH stage, Section 3.3
5:        $\boldsymbol{x}_{\text{search}} \leftarrow$ SEARCHORACLE             ▷ local Bayesian optimization step
6:        Evaluate $f$ on $\boldsymbol{x}_{\text{search}}$, **if** improvement is *sufficient* **then break**
7:     **if** SEARCH is NOT *successful* **then**           ▷ optional POLL stage, Section 3.3
8:        compute poll set $P_k$
9:        evaluate opportunistically $f$ on $P_k$ sorted by acquisition function
10:     **if** iteration $k$ is *successful* **then**
11:        update incumbent $\boldsymbol{x}_{k+1}$
12:        **if** POLL was *successful* **then** $\Delta_k^{\text{mesh}} \leftarrow 2\Delta_k^{\text{mesh}}$, $\Delta_k^{\text{poll}} \leftarrow 2\Delta_k^{\text{poll}}$
13:     **else**
14:        $\Delta_k^{\text{mesh}} \leftarrow \frac{1}{2}\Delta_k^{\text{mesh}}$, $\Delta_k^{\text{poll}} \leftarrow \frac{1}{2}\Delta_k^{\text{poll}}$
15:     $k \leftarrow k + 1$
16: **until** `fevals` > `MaxFunEvals` **or** $\Delta_k^{\text{poll}} < 10^{-6}$ **or** stalling          ▷ stopping criteria
17: **return** $\boldsymbol{x}_{\text{end}} = \arg\min_k f(\boldsymbol{x}_k)$ (or $\boldsymbol{x}_{\text{end}} = \arg\min_k q_\beta(\boldsymbol{x}_k)$ for noisy objectives, Section 3.4)

---

requirement that for each dimension $1 \le d \le D$, $\text{LB}_d \le \text{PLB}_d < \text{PUB}_d \le \text{UB}_d$.[2] Plausible bounds identify a region in parameter space where most solutions are expected to lie. Hard upper/lower bounds can be infinite, but plausible bounds need to be finite. Problem variables whose hard bounds are strictly positive and $\text{UB}_d \ge 10 \cdot \text{LB}_d$ are automatically converted to log space. All variables are then linearly rescaled to the standardized box $[-1, 1]^D$ such that the box bounds correspond to [PLB, PUB] in the original space. BADS supports bound or no constraints, and optionally other constraints via a provided *barrier* function $c$ (see Supplementary Material). The user can also specify circular or periodic dimensions (such as angles); and whether the objective $f$ is deterministic or noisy (stochastic), and in the latter case provide a coarse estimate of the noise (see Section 3.4).

**Initial design**    The initial design consists of the provided starting point $\boldsymbol{x}_0$ and $n_{\text{init}} = D$ additional points chosen via a space-filling quasi-random Sobol sequence [20] in the standardized box, and forced to lie on the mesh grid. If the user does not specify whether $f$ is deterministic or stochastic, the algorithm assesses it by performing two consecutive evaluations at $\boldsymbol{x}_0$.

## 3.2 GP model in BADS

The default GP model is specified by a constant mean function $m \in \mathbb{R}$, a smooth ARD RQ kernel (Eq. 1), and we use $a_{\text{LCB}}$ (Eq. 2) as a default acquisition function.

**Hyperparameters**    The default GP has hyperparameters $\boldsymbol{\theta} = (\ell_1, \ldots, \ell_D, \sigma_f^2, \alpha, \sigma^2, m)$. We impose an empirical Bayes prior on the GP hyperparameters based on the current training set (see Supplementary Material), and select $\boldsymbol{\theta}$ via maximum a posteriori (MAP) estimation. We fit $\boldsymbol{\theta}$ via a gradient-based nonlinear optimizer, starting from either the previous value of $\boldsymbol{\theta}$ or a weighted draw from the prior, as a means to escape local optima. We refit the hyperparameters every $2D$ to $5D$ function evaluations; more often earlier in the optimization, and whenever the current GP is particularly inaccurate at predicting new points, according to a normality test on the residuals, $z^{(i)} = \left(y^{(i)} - \mu(\boldsymbol{x}^{(i)})\right) / \sqrt{s^2(\boldsymbol{x}^{(i)}) + \sigma^2}$ (assumed independent, in first approximation).

**Training set**    The GP training set $\mathbf{X}$ consists of a subset of the points evaluated so far (the *cache*), selected to build a local approximation of the objective in the neighborhood of the incumbent $\boldsymbol{x}_k$, constructed as follows. Each time $\mathbf{X}$ is rebuilt, points in the cache are sorted by their $\boldsymbol{\ell}$-scaled distance $r^2$ (Eq. 1) from $\boldsymbol{x}_k$. First, the closest $n_{\text{min}} = 50$ points are automatically added to $\mathbf{X}$. Second, up to $10D$ additional points with $r \le 3\rho(\alpha)$ are included in the set, where $\rho(\alpha) \gtrsim 1$ is a radius

function that depends on the decay of the kernel. For the RQ kernel, $\rho_{\mathrm{RQ}}(\alpha) = \sqrt{\alpha}\sqrt{e^{1/\alpha} - 1}$ (see Supplementary Material). Newly evaluated points are added incrementally to the set, using fast rank-one updates of the GP posterior. The training set is rebuilt any time the incumbent is moved.

### 3.3   Implementation of the MADS framework

We initialize $\Delta_0^{\mathrm{poll}} = 1$ and $\Delta_0^{\mathrm{mesh}} = 2^{-10}$ (in standardized space), such that the initial poll steps can span the plausible region, whereas the mesh grid is relatively fine. We use $\tau = 2$, and increase the mesh size only after a successful POLL. We skip the POLL after a successful SEARCH.

**Search stage**   We apply an aggressive, repeated SEARCH strategy that consists of up to $n_{\mathrm{search}} = \max\{D, \lfloor 3 + D/2 \rfloor\}$ unsuccessful SEARCH steps. In each step, we use a *search oracle*, based on a local BO with the current GP, to produce a search point $\boldsymbol{x}_{\mathrm{search}}$ (see below). We evaluate $f(\boldsymbol{x}_{\mathrm{search}})$ and add it to the training set. If the improvement in objective value is none or *insufficient*, that is less than $(\Delta_k^{\mathrm{poll}})^{3/2}$, we continue searching, or switch to POLL after $n_{\mathrm{search}}$ steps. Otherwise, we call it a *success* and start a new SEARCH from scratch, centered on the updated incumbent.

**Search oracle**   We choose $\boldsymbol{x}_{\mathrm{search}}$ via a fast, approximate optimization inspired by CMA-ES [21]. We sample batches of points in the neighborhood of the incumbent $\boldsymbol{x}_k$, drawn $\sim \mathcal{N}(\boldsymbol{x}_s, \lambda^2 (\Delta_k^{\mathrm{poll}})^2 \boldsymbol{\Sigma})$, where $\boldsymbol{x}_s$ is the current search focus, $\boldsymbol{\Sigma}$ a *search covariance matrix*, and $\lambda > 0$ a scaling factor, and we pick the point that optimizes the acquisition function (see Supplementary Material). We remove from the SEARCH set candidate points that violate non-bound constraints ($c(\boldsymbol{x}) > 0$), and we project candidate points that fall outside hard bounds to the closest mesh point inside the bounds. Across SEARCH steps, we use both a diagonal matrix $\boldsymbol{\Sigma}_{\boldsymbol{\ell}}$ with diagonal $\left(\ell_1^2/|\boldsymbol{\ell}|^2, \ldots, \ell_D^2/|\boldsymbol{\ell}|^2\right)$, and a matrix $\boldsymbol{\Sigma}_{\mathrm{WCM}}$ proportional to the weighted covariance matrix of points in $\mathbf{X}$ (each point weighted according to a function of its ranking in terms of objective values $y_i$). We choose between $\boldsymbol{\Sigma}_{\boldsymbol{\ell}}$ and $\boldsymbol{\Sigma}_{\mathrm{WCM}}$ probabilistically via a *hedge* strategy, based on their track record of cumulative improvement [22].

**Poll stage**   We incorporate the GP approximation in the POLL in two ways: when constructing the set of polling directions $\mathbf{D}_k$, and when choosing the polling order. We generate $\mathbf{D}_k$ according to the random LTMADS algorithm [6], but then rescale each vector coordinate $1 \leq d \leq D$ proportionally to the GP length scale $\ell_d$ (see Supplementary Material). We discard poll vectors that do not satisfy the given bound or nonbound constraints. Second, since the POLL is opportunistic, we evaluate points in the poll set according to the ranking given by the acquisition function [9].

**Stopping criteria**   We stop the optimization when the poll size $\Delta_k^{\mathrm{poll}}$ goes below a threshold (default $10^{-6}$); when reaching a maximum number of objective evaluations (default $500D$); or if there is no significant improvement of the objective for more than $4 + \lfloor D/2 \rfloor$ iterations. The algorithm returns the optimum $\boldsymbol{x}_{\mathrm{end}}$ (transformed back to original coordinates) with the lowest objective value $y_{\mathrm{end}}$.

### 3.4   Noisy objective

In case of a noisy objective, we assume for the noise a hyperprior $\ln \sigma \sim \mathcal{N}(\ln \sigma_{\mathrm{est}}, 1)$, with $\sigma_{\mathrm{est}}$ a base noise magnitude (default $\sigma_{\mathrm{est}} = 1$, but the user can provide an estimate). To account for additional uncertainty, we also make the following changes: double the minimum number of points added to the training set, $n_{\mathrm{min}} = 100$, and increase the maximum number to 200; increase the initial design to $n_{\mathrm{init}} = 20$; and double the number of allowed stalled iterations before stopping.

**Uncertainty handling**   Due to noise, we cannot simply use the output values $y_i$ as ground truth in the SEARCH and POLL stages. Instead, we replace $y_i$ with the GP latent quantile function [23]

$$q_\beta\left(\boldsymbol{x}; \{\mathbf{X}, \boldsymbol{y}\}, \boldsymbol{\theta}\right) \equiv q_\beta(\boldsymbol{x}) = \mu(\boldsymbol{x}) + \Phi^{-1}(\beta)s(\boldsymbol{x}), \qquad \beta \in [0.5, 1), \tag{3}$$

where $\Phi^{-1}(\cdot)$ is the quantile function of the standard normal (*plugin* approach [24]). Moreover, we modify the MADS procedure by keeping an *incumbent set* $\{\boldsymbol{x}_i\}_{i=1}^k$, where $\boldsymbol{x}_i$ is the incumbent at the end of the $i$-th iteration. At the end of each POLL we re-evaluate $q_\beta$ for all elements of the incumbent set, in light of the new points added to the cache. We select as current (active) incumbent the point with lowest $q_\beta(\boldsymbol{x}_i)$. During optimization we set $\beta = 0.5$ (mean prediction only), which promotes exploration. We use a conservative $\beta_{\mathrm{end}} = 0.999$ for the last iteration, to select the optimum $\boldsymbol{x}_{\mathrm{end}}$ returned by the algorithm in a robust manner. Instead of $y_{\mathrm{end}}$, we return either $\mu(\boldsymbol{x}_{\mathrm{end}})$ or an unbiased estimate of $\mathbb{E}[f(\boldsymbol{x}_{\mathrm{end}})]$ obtained by averaging multiple evaluations (see Supplementary Material).

## 4    Experiments

We tested BADS and many optimizers with implementation available in MATLAB (R2015b, R2017a) on a large set of artificial and real optimization problems (see Supplementary Material for details).

### 4.1    Design of the benchmark

**Algorithms**    Besides BADS, we tested 16 optimization algorithms, including popular choices such as Nelder-Mead (`fminsearch` [25]), several constrained nonlinear optimizers in the `fmincon` function (default *interior-point* [26], *sequential quadratic programming* `sqp` [27], and *active-set* `actset` [28]), genetic algorithms (`ga` [29]), random search (`randsearch`) as a baseline [30]; and also less-known state-of-the-art methods for nonconvex derivative-free optimization [1], such as Multilevel Coordinate Search (MCS [31]) and CMA-ES [21, 32] (`cmaes`, in different flavors). For noisy objectives, we included algorithms that explicitly handle uncertainty, such as `snobfit` [33] and *noisy* CMA-ES [34]. Finally, to verify the advantage of BADS' hybrid approach to BO, we also tested a standard, 'vanilla' version of BO [5] (`bayesopt`, R2017a) on the set of real model-fitting problems (see below). For all algorithms, including BADS, we used default settings (no fine-tuning).

**Problem sets**    First, we considered a standard benchmark set of artificial, noiseless functions (BBOB09 [35], 24 functions) in dimensions $D \in \{3, 6, 10, 15\}$, for a total of 96 test functions. We also created 'noisy' versions of the same set. Second, we collected model-fitting problems from six published or ongoing studies in cognitive and computational neuroscience (CCN17). The objectives of the CCN17 set are negative log likelihood functions of an input parameter vector, for specified datasets and models, and can be deterministic or stochastic. For each study in the CCN17 set we asked its authors for six different real datasets (i.e., subjects or neurons), divided between one or two main models of interest; collecting a total of 36 test functions with $D \in \{6, 9, 10, 12, 13\}$.

**Procedure**    We ran 50 independent runs of each algorithm on each test function, with randomized starting points and a budget of $500 \times D$ function evaluations ($200 \times D$ for noisy problems). If an algorithm terminated before depleting the budget, it was restarted from a new random point. We consider a run *successful* if the current best (or returned, for noisy problems) function value is within a given *error tolerance* $\varepsilon > 0$ from the true optimum $f_{\min}$ (or our best estimate thereof).[3] For noiseless problems, we compute the fraction of successful runs as a function of number of objective evaluations, averaged over datasets/functions and over $\varepsilon \in [0.01, 10]$ (log spaced). This is a realistic range for $\varepsilon$, as differences in log likelihood below 0.01 are irrelevant for model selection; an acceptable tolerance is $\varepsilon \sim 0.5$ (a difference in *deviance*, the metric used for AIC or BIC, less than 1); larger $\varepsilon$ associate with coarse solutions, but errors larger than 10 would induce excessive biases in model selection. For noisy problems, what matters most is the solution $x_{\mathrm{end}}$ that the algorithm *actually* returns, which, depending on the algorithm, may not necessarily be the point with the lowest *observed* function value. Since, unlike the noiseless case, we generally do not know the solutions that would be returned by any algorithm at every time step, but only at the last step, we plot instead the fraction of successful runs at $200 \times D$ function evaluations as a function of $\varepsilon$, for $\varepsilon \in [0.1, 10]$ (noise makes higher precisions moot), and averaged over datasets/functions. In all plots we omit error bars for clarity (standard errors would be about the size of the line markers or less).

### 4.2    Results on artificial functions (BBOB09)

The BBOB09 noiseless set [35] comprises of 24 functions divided in 5 groups with different properties: separable; low or moderate conditioning; unimodal with high conditioning; multi-modal with adequate / with weak global structure. First, we use this benchmark to show the performance of different configurations for BADS. Note that we selected the default configuration (RQ kernel, $a_{\mathrm{LCB}}$) and other algorithmic details by testing on a different benchmark set (see Supplementary Material). Fig 1 (left) shows aggregate results across all noiseless functions with $D \in \{3, 6, 10, 15\}$, for alternative choices of kernels and acquisition functions (only a subset is shown, such as the popular $M_{5/2}$, EI combination), or by altering other features (such as setting $n_{\mathrm{search}} = 1$, or fixing the search covariance matrix to $\Sigma_{\ell}$ *or* $\Sigma_{\mathrm{WCM}}$). Almost all changes from the default configuration worsen performance.

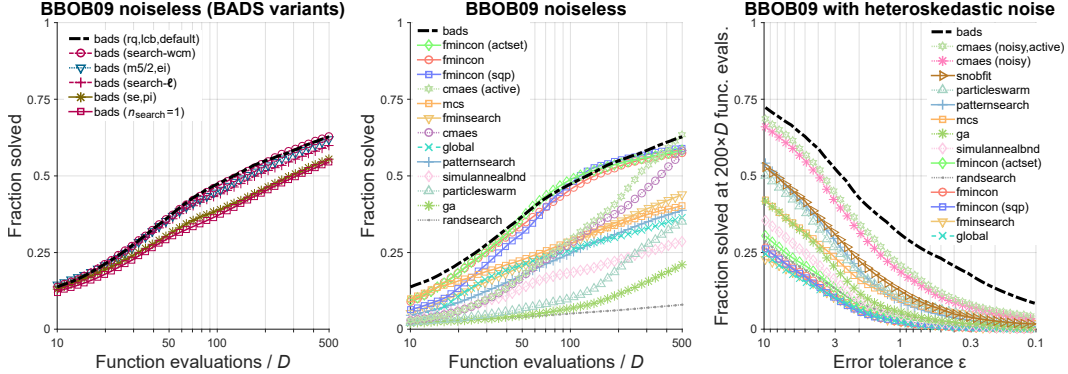

Figure 1: **Artificial test functions (BBOB09).** *Left & middle*: Noiseless functions. Fraction of successful runs ($\varepsilon \in [0.01, 10]$) vs. # function evaluations per # dimensions, for $D \in \{3, 6, 10, 15\}$ (96 test functions); for different BADS configurations (*left*) and all algorithms (*middle*). *Right*: Heteroskedastic noise. Fraction of successful runs at $200 \times D$ objective evaluations vs. tolerance $\varepsilon$.

**Noiseless functions** We then compared BADS to other algorithms (Fig 1 middle). Depending on the number of function evaluations, the best optimizers are BADS, methods of the `fmincon` family, and, for large budget of function evaluations, CMA-ES with *active* update of the covariance matrix.

**Noisy functions** We produce noisy versions of the BBOB09 set by adding i.i.d. Gaussian observation noise at each function evaluation, $y^{(i)} = f(\boldsymbol{x}^{(i)}) + \sigma(\boldsymbol{x}^{(i)})\eta^{(i)}$, with $\eta^{(i)} \sim \mathcal{N}(0, 1)$. We consider a variant with moderate *homoskedastic* (constant) noise ($\sigma = 1$), and a variant with *heteroskedastic* noise with $\sigma(\boldsymbol{x}) = 1 + 0.1 \times (f(\boldsymbol{x}) - f_{\min})$, which follows the observation that variability generally increases for solutions away from the optimum. For many functions in the BBOB09 set, this heteroskedastic noise can become substantial ($\sigma \gg 10$) away from the optimum. Fig 1 (right) shows aggregate results for the heteroskedastic set (homoskedastic results are similar). BADS outperforms all other optimizers, with CMA-ES (*active*, with or without the *noisy* option) coming second.

Notably, BADS performs well even on problems with non-stationary (location-dependent) features, such as heteroskedastic noise, thanks to its local GP approximation.

## 4.3 Results on real model-fitting problems (CCN17)

The objectives of the CCN17 set are deterministic (e.g., computed via numerical approximation) for three studies (Fig 2), and noisy (e.g., evaluated via simulation) for the other three (Fig 3).

The algorithmic cost of BADS is $\sim 0.03$ s to $0.15$ s per function evaluation, depending on $D$, mostly due to the refitting of the GP hyperparameters. This produces a non-negligible *overhead*, defined as $100\% \times$ (*total optimization time / total function time* $-1$). For a fair comparison with other methods with little or no overhead, for deterministic problems we also plot the *effective* performance of BADS by accounting for the extra cost per function evaluation. In practice, this correction shifts rightward the performance curve of BADS in log-iteration space, since each function evaluation with BADS has an increased fractional time cost. For stochastic problems, we cannot compute effective performance as easily, but there we found small overheads ($< 5\%$), due to more costly evaluations (more than 1 s).

For a direct comparison with standard BO, we also tested on the CCN17 set a 'vanilla' BO algorithm, as implemented in MATLAB R2017a (`bayesopt`). This implementation closely follows [5], with optimization instead of marginalization over GP hyperparameters. Due to the fast-growing cost of BO as a function of training set size, we allowed up to 300 training points for the GP, restarting the BO algorithm from scratch with a different initial design every 300 BO iterations (until the total budget of function evaluations was exhausted). The choice of 300 iterations already produced a large average algorithmic overhead of $\sim 8$ s per function evaluation. In showing the results of `bayesopt`, we display raw performance without penalizing for the overhead.

**Causal inference in visuo-vestibular perception** Causal inference (CI) in perception is the process whereby the brain decides whether to integrate or segregate multisensory cues that could arise from the same or from different sources [39]. This study investigates CI in visuo-vestibular heading

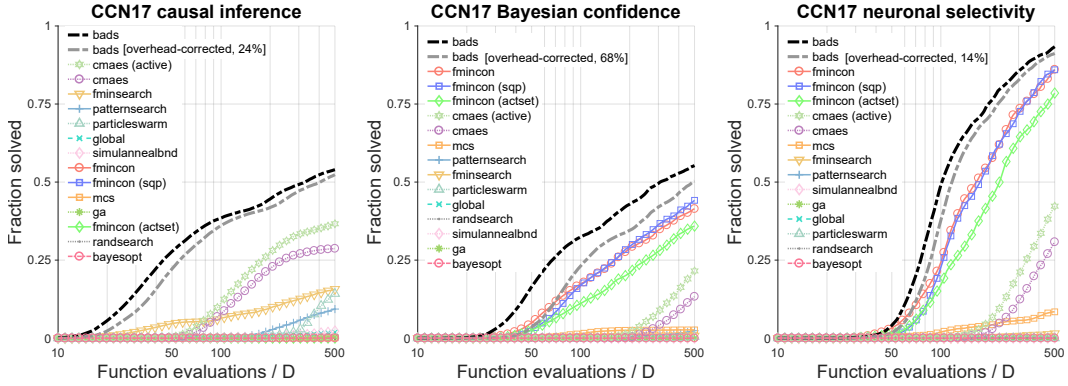

Figure 2: **Real model-fitting problems (CCN17, deterministic).** Fraction of successful runs ($\varepsilon \in [0.01, 10]$) vs. # function evaluations per # dimensions. *Left*: Causal inference in visuo-vestibular perception [36] (6 subjects, $D = 10$). *Middle*: Bayesian confidence in perceptual categorization [37] (6 subjects, $D = 13$). *Right*: Neural model of orientation selectivity [38] (6 neurons, $D = 12$).

perception across tasks and under different levels of visual reliability, via a factorial model comparison [36]. For our benchmark we fit three subjects with a Bayesian CI model ($D = 10$), and another three with a fixed-criterion CI model ($D = 10$) that disregards visual reliability. Both models include heading-dependent likelihoods and marginalization of the decision variable over the latent space of noisy sensory measurements ($x_{\text{vis}}, x_{\text{vest}}$), solved via nested numerical integration in 1-D and 2-D.

**Bayesian confidence in perceptual categorization**  This study investigates the *Bayesian confidence hypothesis* that subjective judgments of confidence are directly related to the posterior probability the observer assigns to a learnt perceptual category [37] (e.g., whether the orientation of a drifting Gabor patch belongs to a 'narrow' or to a 'wide' category). For our benchmark we fit six subjects to the 'Ultrastrong' Bayesian confidence model ($D = 13$), which uses the same mapping between posterior probability and confidence across two tasks with different distributions of stimuli. This model includes a latent noisy decision variable, marginalized over via 1-D numerical integration.

**Neural model of orientation selectivity**  The authors of this study explore the origins of diversity of neuronal orientation selectivity in visual cortex via novel stimuli (orientation mixtures) and modeling [38]. We fit the responses of five V1 and one V2 cells with the authors' neuronal model ($D = 12$) that combines effects of filtering, suppression, and response nonlinearity [38]. The model has one circular parameter, the preferred direction of motion of the neuron. The model is analytical but still computationally expensive due to large datasets and a cascade of several nonlinear operations.

**Word recognition memory**  This study models a word recognition task in which subjects rated their confidence that a presented word was in a previously studied list [40] (data from [41]). We consider six subjects divided between two normative models, the 'Retrieving Effectively from Memory' model [42] ($D = 9$) and a similar, novel model[4] ($D = 6$). Both models use Monte Carlo methods to draw random samples from a large space of latent noisy memories, yielding a stochastic log likelihood.

**Target detection and localization**  This study looks at differences in observers' decision making strategies in target detection ('was the target present?') and localization ('which one was the target?') with displays of $2, 3, 4$, or $6$ oriented Gabor patches.[5] Here we fit six subjects with a previously derived ideal observer model [43, 44] ($D = 6$) with variable-precision noise [45], assuming shared parameters between detection and localization. The log likelihood is evaluated via simulation due to marginalization over latent noisy measurements of stimuli orientations with variable precision.

**Combinatorial board game playing**  This study analyzes people's strategies in a four-in-a-row game played on a 4-by-9 board against human opponents ([46], Experiment 1). We fit the data of six players with the *main* model ($D = 10$), which is based on a Best-First exploration of a decision tree guided by a feature-based value heuristic. The model also includes feature dropping, value noise, and lapses, to better capture human variability. Model evaluation is computationally expensive due to the

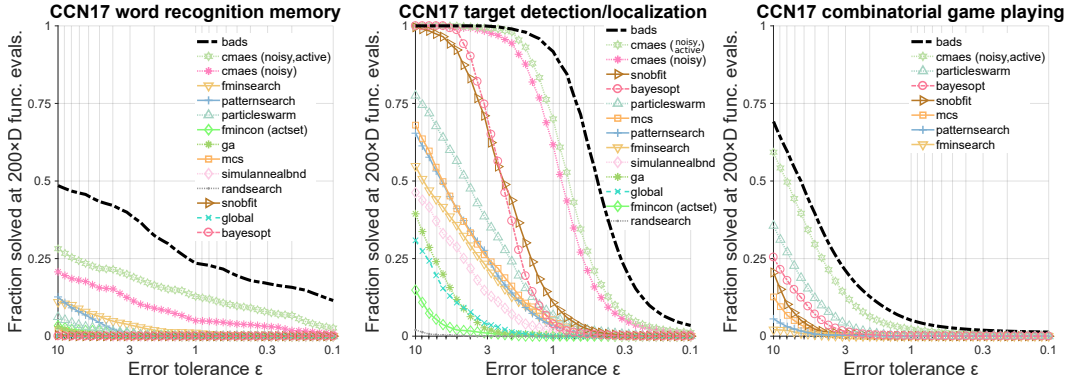

Figure 3: **Real model-fitting problems (CCN17, noisy).** Fraction of successful runs at $200 \times D$ objective evaluations vs. tolerance $\varepsilon$. *Left*: Confidence in word recognition memory [40] (6 subjects, $D = 6, 9$). *Middle*: Target detection and localization [44] (6 subjects, $D = 6$). *Right*: Combinatorial board game playing [46] (6 subjects, $D = 10$).

construction and evaluation of trees of future board states, and achieved via *inverse binomial sampling*, an unbiased stochastic estimator of the log likelihood [46]. Due to prohibitive computational costs, here we only test major algorithms (MCS is the method used in the paper [46]); see Fig 3 right.

In all problems, BADS consistently performs on par with or outperforms all other tested optimizers, even when accounting for its extra algorithmic cost. The second best algorithm is either some flavor of CMA-ES or, for some deterministic problems, a member of the fmincon family. Crucially, their ranking across problems is inconsistent, with both CMA-ES and fmincon performing occasionally quite poorly (e.g., fmincon does poorly in the *causal inference* set because of small fluctuations in the log likelihood landscape caused by coarse numerical integration). Interestingly, vanilla BO (bayesopt) performs poorly on all problems, often at the level of random search, and always substantially worse than BADS, even without accounting for the much larger overhead of bayesopt. The solutions found by bayesopt are often hundreds (even thousands) points of log likelihood from the optimum. This failure is possibly due to the difficulty of building a *global* GP surrogate for BO, coupled with strong non-stationarity of the log likelihood functions; and might be ameliorated by more complex forms of BO (e.g., input warping to produce nonstationary kernels [47], hyperparameter marginalization [5]). However, these advanced approaches would substantially increase the already large overhead. Importantly, we expect this poor perfomance to extend to any package which implements vanilla BO (such as *BayesOpt* [48]), regardless of the efficiency of implementation.

## 5 Conclusions

We have developed a novel BO method and an associated toolbox, BADS, with the goal of fitting moderately expensive computational models out-of-the-box. We have shown on real model-fitting problems that BADS outperforms widely used and state-of-the-art methods for nonconvex, derivative-free optimization, including 'vanilla' BO. We attribute the robust performance of BADS to the alternation between the aggressive SEARCH strategy, based on local BO, and the failsafe POLL stage, which protects against failures of the GP surrogate – whereas vanilla BO does not have such failsafe mechanisms, and can be strongly affected by model misspecification. Our results demonstrate that a hybrid Bayesian approach to optimization can be beneficial beyond the domain of very costly black-box functions, in line with recent advancements in probabilistic numerics [49].

Like other surrogate-based methods, the performance of BADS is linked to its ability to obtain a fast approximation of the objective, which generally deteriorates in high dimensions, or for functions with pathological structure (often improvable via reparameterization). From our tests, we recommend BADS, paired with some multi-start optimization strategy, for models with up to $\sim 15$ variables, a noisy or jagged log likelihood landscape, and when algorithmic overhead is $\lesssim 75\%$ (e.g., model evaluation $\gtrsim 0.1$ s). Future work with BADS will focus on testing alternative statistical surrogates instead of GPs [12]; combining it with a smart multi-start method for global optimization; providing support for tunable precision of noisy observations [23]; improving the numerical implementation; and recasting some of its heuristics in terms of approximate inference.

## Acknowledgments

We thank Will Adler, Robbe Goris, Andra Mihali, Bas van Opheusden, and Aspen Yoo for sharing data and model evaluation code that we used in the CCN17 benchmark set; Maija Honig, Andra Mihali, Bas van Opheusden, and Aspen Yoo for providing user feedback on earlier versions of the `bads` package for MATLAB; Will Adler, Andra Mihali, Bas van Opheusden, and Aspen Yoo for helpful feedback on a previous version of this manuscript; John Wixted and colleagues for allowing us to reuse their data for the CCN17 'word recognition memory' problem set; and the three anonymous reviewers for useful feedback. This work has utilized the NYU IT High Performance Computing resources and services.

## Footnotes

[1]Code available at `https://github.com/lacerbi/bads`.

[2] A variable $d$ can be *fixed* by setting $(\boldsymbol{x}_0)_d = \text{LB}_d = \text{UB}_d = \text{PLB}_d = \text{PUB}_d$. Fixed variables become constants, and BADS runs on an optimization problem with reduced dimensionality.

[3]Note that the error tolerance $\varepsilon$ is *not* a fractional error, as sometimes reported in optimization, because for model comparison we typically care about (absolute) differences in log likelihoods.

[4]Unpublished; upcoming work from Aspen H. Yoo and Wei Ji Ma.

[5]Unpublished; upcoming work from Andra Mihali and Wei Ji Ma.

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
