[Supplementary Material · bads_supplement.pdf]

# Practical Bayesian Optimization for Model Fitting with Bayesian Adaptive Direct Search – Supplementary Material

**Luigi Acerbi**[*]
Center for Neural Science
New York University
luigi.acerbi@nyu.edu

**Wei Ji Ma**
Center for Neural Science & Dept. of Psychology
New York University
weijima@nyu.edu

In this Supplement, we expand on the definitions and implementations of Gaussian Processes (GPs) and Bayesian optimization in BADS (Section A); we give a full description of the BADS algorithm, including details omitted in the main text (Section B); we report further details of the benchmark procedure, such as the full list of tested algorithms and additional results (Section C); and, finally, we briefly discuss the numerical implementation (Section D).

## A    Gaussian processes for Bayesian optimization in BADS

In this section, we describe definitions and additional specifications of the Gaussian process (GP) model used for Bayesian optimization (BO) in BADS. Specifically, this part expands on Sections 2.2 and 3.2 in the main text.

**GP posterior moments**    We consider a GP based on a training set $\mathbf{X}$ with $n$ points, a vector of observed function values $\boldsymbol{y}$, and GP mean function $m(\boldsymbol{x})$ and GP covariance or kernel function $k(\boldsymbol{x}, \boldsymbol{x}')$, with i.i.d. Gaussian observation noise $\sigma^2 > 0$. The GP posterior latent marginal conditional mean $\mu$ and variance $s^2$ are available in closed form at a chosen point as

$$
\begin{aligned}
\mu\left(\boldsymbol{x}\right) &\equiv \mu\left(\boldsymbol{x}; \{\mathbf{X}, \boldsymbol{y}\}, \boldsymbol{\theta}\right) = \boldsymbol{k}(\boldsymbol{x})^{\top}\left(\mathbf{K} + \sigma^2 \mathbf{I}_n\right)^{-1}\left(\boldsymbol{y} - m(\boldsymbol{x})\right) \\
s^2\left(\boldsymbol{x}\right) &\equiv s^2\left(\boldsymbol{x}; \{\mathbf{X}, \boldsymbol{y}\}, \boldsymbol{\theta}\right) = k(\boldsymbol{x}, \boldsymbol{x}) - \boldsymbol{k}(\boldsymbol{x})^{\top}\left(\mathbf{K} + \sigma^2 \mathbf{I}_n\right)^{-1}\boldsymbol{k}(\boldsymbol{x})
\end{aligned}
\tag{S1}
$$

where $\mathbf{K}_{ij} = k(\boldsymbol{x}^{(i)}, \boldsymbol{x}^{(j)})$, for $1 \leq i, j \leq n$, is the kernel matrix, $\boldsymbol{k}(\boldsymbol{x}) \equiv (k(\boldsymbol{x}, \boldsymbol{x}^{(1)}), \ldots, k(\boldsymbol{x}, \boldsymbol{x}^{(n)}))^{\top}$ is the $n$-dimensional column vector of cross-covariances, and $\boldsymbol{\theta}$ is the vector of GP hyperparameters.

### A.1    Covariance functions

Besides the automatic relevance determination (ARD) *rational quadratic* (RQ) kernel described in the main text (and BADS default), we also considered the common *squared exponential* (SE) kernel

$$
k_{\text{SE}}\left(\boldsymbol{x}, \boldsymbol{x}'\right) = \sigma_f^2 \exp\left\{-\frac{1}{2} r^2(\boldsymbol{x}, \boldsymbol{x}')\right\}, \qquad \text{with } r^2(\boldsymbol{x}, \boldsymbol{x}') = \sum_{d=1}^{D} \frac{1}{\ell_d^2}\left(x_d - x_d'\right)^2,
\tag{S2}
$$

and the ARD *Matérn 5/2* kernel [1],

$$
k_{\text{M52}}\left(\boldsymbol{x}, \boldsymbol{x}'\right) = \sigma_f^2 \left[1 + \sqrt{5 r^2(\boldsymbol{x}, \boldsymbol{x}')} + \frac{5}{3} r^2(\boldsymbol{x}, \boldsymbol{x}')\right] \exp\left\{-\sqrt{5 r^2(\boldsymbol{x}, \boldsymbol{x}')}\right\},
\tag{S3}
$$

where $\sigma_f^2$ is the signal variance, and $\ell_1, \ldots, \ell_D$ are the kernel length scales along each coordinate. Note that the RQ kernel tends to the SE kernel for $\alpha \to \infty$.

The Matérn 5/2 kernel has become a more common choice for Bayesian *global* optimization because it is only twice-differentiable [1], whereas the SE and RQ kernels are infinitely differentiable – a

---

[*]Current address: Département des neurosciences fondamentales, Université de Genève, CMU, 1 rue Michel-Servet, 1206 Genève, Switzerland. E-mail: luigi.acerbi@gmail.com.

stronger assumption of smoothness which may cause extrapolation issues. However, this is less of a problem for a local interpolating approximation (as in BADS) than it is for a global approach, and in fact we find the RQ kernel to work well empirically (see main text).

**Composite periodic kernels** We allow the user to specify one or more *periodic* (equivalently, circular) coordinate dimensions $P \subseteq \{1, \ldots, D\}$, which is a feature of some models in computational neuroscience (e.g., the preferred orientation of a neuron, as in the 'neuronal selectivity' problem set [2] of the CCN17 benchmark; see Section 4.3 in the main text). For a chosen base stationary covariance function $k_0$ (e.g., RQ, SE, $M_{5/2}$), we define the composite ARD periodic kernel as

$$k_{\text{PER}}(\boldsymbol{x}, \boldsymbol{x}'; k_0, P) = k_0\left(t(\boldsymbol{x}), t(\boldsymbol{x}')\right), \quad \text{with} \quad \begin{cases} [t(\boldsymbol{x})]_d &= x_d & \text{if } d \notin P \\ [t(\boldsymbol{x})]_d &= \sin\left(\frac{\pi x_d}{L_d}\right) & \text{if } d \in P \\ [t(\boldsymbol{x})]_{d+|P|} &= \cos\left(\frac{\pi x_d}{L_d}\right) & \text{if } d \in P \end{cases} \quad \text{(S4)}$$

for $1 \leq d \leq D$, where $L_d$ is the period in the $d$-th coordinate dimension, and the length scale $\ell_d$ of $k_0$ is shared between $(d, d + |P|)$ pairs when $d \in P$. In BADS, the period is determined by the provided hard bounds as $L_d = \text{UB}_d - \text{LB}_d$ (where the hard bounds are required to be finite).

## A.2 Construction of the training set

We construct the training set $\mathbf{X}$ according to a simple *subset-of-data* [3] local GP approximation. Points are added to the training set sorted by their $\boldsymbol{\ell}$-scaled distance $r^2$ from the incumbent $\boldsymbol{x}_k$. The training set contains a minimum of $n_{\min} = 50$ points (if available in the cache of all points evaluated so far), and then up to $10 \times D$ additional points with $r \leq 3\rho(\alpha)$, where $\rho(\alpha)$ is a *radius function* that depends on the decay of the kernel. For a given stationary kernel of the form $k(\boldsymbol{x}, \boldsymbol{x}') = k(r^2(\boldsymbol{x}, \boldsymbol{x}'))$, we define $\rho$ as the distance such that $k(2\rho^2) \equiv 1/(\sigma_f^2 e)$. We have then

$$\rho_{SE} = 1, \qquad \rho_{M52} \approx 0.92, \quad \text{and} \quad \rho_{RQ}(\alpha) = \sqrt{\alpha(e^{1/\alpha} - 1)}, \quad \text{(S5)}$$

where for example $\rho_{RQ}(1) \approx 1.31$, and $\lim_{\alpha \to \infty} \rho_{RQ}(\alpha) = 1$.

## A.3 Treatment of hyperparameters

We fit the GP hyperparameters by maximizing their posterior probability (MAP), $p(\boldsymbol{\theta}|\mathbf{X}, \boldsymbol{y}) \propto p(\boldsymbol{\theta}, \mathbf{X}, \boldsymbol{y})$, which, thanks to the Gaussian likelihood, is available in closed form as [4]

$$\ln p(\boldsymbol{y}, \mathbf{X}, \boldsymbol{\theta}) = -\frac{1}{2} \ln |\mathbf{K} + \sigma^2 \mathbf{I}_n| - \frac{1}{2} \boldsymbol{y}^\top \left(\mathbf{K} + \sigma^2 \mathbf{I}_n\right)^{-1} \boldsymbol{y} + \ln p_{\text{hyp}}(\boldsymbol{\theta}) + \text{const}, \quad \text{(S6)}$$

where $\mathbf{I}_n$ is the identity matrix in dimension $n$ (the number of points in the training set), and $p_{\text{hyp}}(\boldsymbol{\theta})$ is the prior over hyperparameters, described in the following.

**Hyperparameter prior** We adopt an approximate *empirical Bayes* approach by defining the prior based on the data in the training set, that is $p_{\text{hyp}} = p_{\text{hyp}}(\boldsymbol{\theta}; \mathbf{X}, \boldsymbol{y})$. Empirical Bayes can be intended as a quick, heuristic approximation to a proper but more expensive hierarchical Bayesian approach. We assume independent priors for each hyperparameter, with bounded (truncated) distributions. Hyperparameter priors and hard bounds are reported in Table S1. In BADS, we include an observation noise parameter $\sigma > 0$ also for deterministic objectives $f$, merely for the purpose of fitting the GP, since it has been shown to yield several advantages [5]. In particular, we assume a prior such that $\sigma$ decreases as a function of the poll size $\Delta_k^{\text{poll}}$, as the optimization 'zooms in' to smaller scales. Another distinctive choice for BADS is that we set the mean for the GP mean equal to the 90-th percentile of the observed values in the current training set $\boldsymbol{y}$, which encourages the exploration to remain local.

**Hyperparameter optimization** We optimize Eq. S6 with a gradient-based optimizer (see Section D), providing the analytical gradient to the algorithm. We start the optimization from the previous hyparameter values $\boldsymbol{\theta}_{\text{prev}}$. If the optimization seems to be stuck in a high-noise mode, or we find an unusually low value for the GP mean $m$, we attempt a second fit starting from a draw from the prior averaged with $\boldsymbol{\theta}_{\text{prev}}$. If the optimization fails due to numerical issues, we keep the previous value of

| Hyperparameter | Prior | Bounds |
|---|---|---|
| **GP kernel** | | |
| Length scales $\ell_d$ | $\ln \ell_d \sim \mathcal{N}_{\mathrm{T}}\left(\frac{1}{2}(\ln r_{\max} + \ln r_{\min}), \frac{1}{4}(\ln r_{\max} - \ln r_{\min})^2\right)$ | $[\Delta_{\min}^{\mathrm{poll}}, L_d]$ |
| Signal variability $\sigma_f$ | $\ln \sigma_f \sim \mathcal{N}_{\mathrm{T}}\left(\ln \mathrm{SD}(\boldsymbol{y}), 2^2\right)$ | $[10^{-3}, 10^9]$ |
| RQ kernel shape $\alpha$ | $\ln \alpha \sim \mathcal{N}_{\mathrm{T}}\left(1, 1\right)$ | $[-5, 5]$ |
| **GP observation noise** $\sigma$ | $\ln \sigma \sim \mathcal{N}_{\mathrm{T}}\left(\ln \sigma_{\mathrm{est}}, 1\right)$ | $[4 \cdot 10^{-4}, 150]$ |
| deterministic $f$ | $\sigma_{\mathrm{est}} = \sqrt{10^{-3}\Delta_k^{\mathrm{poll}}}$ | |
| noisy $f$ | $\sigma_{\mathrm{est}} = 1$ (or user-provided estimate) | |
| **GP mean** $m$ | $m \sim \mathcal{N}\left(\mathrm{Q}_{0.9}(\boldsymbol{y}), \frac{1}{5^2}(\mathrm{Q}_{0.9}(\boldsymbol{y}) - \mathrm{Q}_{0.5}(\boldsymbol{y}))^2\right)$ | $(-\infty, \infty)$ |

Table S1: **GP hyperparameter priors.** Empirical Bayes priors and bounds for GP hyperparameters. $\mathcal{N}\left(\mu, \sigma^2\right)$ denotes the normal pdf with mean $\mu$ and variance $\sigma^2$, and $\mathcal{N}_{\mathrm{T}}\left(\cdot, \cdot\right)$ the *truncated* normal, defined within the bounds specified in the last column. $r_{\max}$ and $r_{\min}$ are the maximum (resp., minimum) distance between any two points in the training set; $\Delta_{\min}^{\mathrm{poll}}$ is the minimum poll size (default $10^{-6}$); $L_d$ is the parameter range ($\mathrm{UB}_d - \mathrm{LB}_d$), for $1 \leq d \leq D$; $\mathrm{SD}(\cdot)$ denotes the standard deviation of a set of elements; $\Delta_k^{\mathrm{poll}}$ is the poll size parameter at the current iteration $k$; $\mathrm{Q}_q(\cdot)$ denotes the $q$-th quantile of a set of elements ($\mathrm{Q}_{0.5}$ is the median).

the hyperparameters. We refit the hyperparameters every $2D$ to $5D$ function evaluations; more often earlier in the optimization, and whenever the current GP is particularly inaccurate at predicting new points. We test accuracy on newly evaluated points via a Shapiro-Wilk normality test on the residuals [6], $z^{(i)} = \left(y^{(i)} - \mu(\boldsymbol{x}^{(i)})\right) / \sqrt{s^2(\boldsymbol{x}^{(i)}) + \sigma^2}$ (assumed independent, in first approximation), and flag the approximation as inaccurate if $p < 10^{-6}$.

## A.4 Acquisition functions

Besides the *GP lower confidence bound* (LCB) metric [7] described in the main text (and default in BADS), we consider two other choices that are available in closed form using Eq. S1 for the GP predictive mean and variance.

**Probability of improvement (PI)** This strategy maximizes the probability of improving over the current best minimum $y_{\mathrm{best}}$ [8]. For consistency with the main text, we define here the *negative* PI,

$$a_{\mathrm{PI}}\left(\boldsymbol{x}; \{\mathbf{X}_n, \boldsymbol{y}_n\}, \boldsymbol{\theta}\right) = -\Phi\left(\gamma(\boldsymbol{x})\right), \qquad \gamma(\boldsymbol{x}) = \frac{y_{\mathrm{best}} - \xi - \mu(\boldsymbol{x})}{s(\boldsymbol{x})} \tag{S7}$$

where $\xi \geq 0$ is an optional trade-off parameter to promote exploration, and $\Phi(\cdot)$ is the cumulative distribution function of the standard normal. $a_{\mathrm{PI}}$ is known to excessively favor exploitation over exploration, and it is difficult to find a correct setting for $\xi$ to offset this tendency [9].

**Expected improvement (EI)** We then consider the popular predicted improvement criterion [1, 10, 11]. The expected improvement over the current best minimum $y_{\mathrm{best}}$ (with an offset $\xi \geq 0$) is defined as $\mathbb{E}\left[\max\left\{y_{\mathrm{best}} - y, 0\right\}\right]$. For consistency with the main text we consider the *negative* EI, which can be computed in closed form as

$$a_{\mathrm{EI}}\left(\boldsymbol{x}; \{\mathbf{X}, \boldsymbol{y}\}, \boldsymbol{\theta}\right) = -s(\boldsymbol{x})\left[\gamma(\boldsymbol{x})\Phi\left(\gamma(\boldsymbol{x})\right) + \mathcal{N}\left(\gamma(\boldsymbol{x})\right)\right] \tag{S8}$$

where $\mathcal{N}(\cdot)$ is the standard normal pdf.

## B  The BADS algorithm

We report here extended details of the BADS algorithm, and how the various steps of the MADS framework are implemented (expanding on Sections 3.1 and 3.3 of the main text). Main features of the algorithm are summarized in Table S2. Refer also to Algorithm 1 in the main text.

| Feature | Description (defaults) |
|---|---|
| Surrogate model | GP |
| Hyperparameter treatment | optimization |
| GP training set size $n_{\max}$ | 70 ($D = 2$), 250 ($D = 20$) (min 200 for noisy problems) |
| POLL directions generation | LTMADS with GP rescaling |
| SEARCH set generation | Two-step ES algorithm with search matrix $\boldsymbol{\Sigma}$ |
| SEARCH evals. ($n_{\text{search}}$) | $\max\{D, 3 + \lfloor D/2 \rfloor\}$ |
| Aquisition function | LCB |
| Supported constraints | None, bound, and non-bound via a barrier function $c$ |
| Initial mesh size | $\Delta_0^{\text{mesh}} = 2^{-10}, \Delta_k^{\text{poll}} = 1$ |
| Implementation | `bads` (MATLAB) |

Table S2: **Summary of features of BADS.**

## B.1 Problem definition and initialization

BADS solves the optimization problem

$$f_{\min} = \min_{x \in \mathcal{X}} f(\boldsymbol{x}) \qquad \text{with} \quad \mathcal{X} \subseteq \mathbb{R}^D$$

(*optional*) $\quad c(\boldsymbol{x}) \leq 0$

(S9)

where $\mathcal{X}$ is defined by pairs of hard bound constraints for each coordinate, $\texttt{LB}_d \leq x_d \leq \texttt{UB}_d$ for $1 \leq d \leq D$, and we allow $\texttt{LB}_d \in \mathbb{R} \cup \{-\infty\}$ and similarly $\texttt{UB}_d \in \mathbb{R} \cup \{\infty\}$. We also consider optional non-bound constraints specified via a *barrier* function $c : \mathcal{X} \to \mathbb{R}$ that returns constraint *violations*. We only consider solutions such that $c$ is zero or less.

**Algorithm input** The algorithm takes as input a starting point $\boldsymbol{x}_0 \in \mathcal{X}$; vectors of *hard* lower/upper bounds LB, UB; optional vectors of *plausible* lower/upper bounds PLB, PUB; and an optional barrier function $c$. We require that, if specified, $c(\boldsymbol{x}_0) \leq 0$; and for each dimension $1 \leq d \leq D$, $\texttt{LB}_d \leq (\boldsymbol{x}_0)_d \leq \texttt{UB}_d$ and $\texttt{LB}_d \leq \texttt{PLB}_d < \texttt{PUB}_d \leq \texttt{UB}_d$. Plausible bounds identify a region in parameter space where most solutions are expected to lie, which in practice we usually think of as the region where starting points for the algorithm would be drawn from. Hard upper/lower bounds can be infinite, but plausible bounds need to be finite. As an exception to the above bound ordering, the user can specify that a variable is *fixed* by setting $(\boldsymbol{x}_0)_d = \texttt{LB}_d = \texttt{UB}_d = \texttt{PLB}_d = \texttt{PUB}_d$. Fixed variables become constants, and BADS runs on an optimization problem with reduced dimensionality. The user can also specify circular or periodic dimensions (such as angles), which change the definition of the GP kernel as per Section A.1. The user can specify whether the objective $f$ is deterministic or noisy (stochastic), and in the latter case provide a coarse estimate of the noise (see Section B.5).

**Transformation of variables and constraints** Problem variables whose hard bounds are strictly positive and $\texttt{UB}_d \geq 10 \cdot \texttt{LB}_d$ are automatically converted to log space for all internal calculations of the algorithm. All variables are also linearly rescaled to the standardized box $[-1, 1]^D$ such that the box bounds correspond to [PLB, PUB] in the original space. BADS converts points back to the original coordinate space when calling the target function $f$ or the barrier function $c$, and at the end of the optimization. BADS never violates constraints, by removing from the POLL and SEARCH sets points that violate either bound or non-bound constraints ($c(\boldsymbol{x}) > 0$). During the SEARCH stage, we project candidate points that violate a bound constraint to the closest mesh point within the bounds. We assume that $c(\cdot)$, if provided, is known and inexpensive to evaluate.

**Objective scaling** We assume that the scale of interest for differences in the objective (and the scale of other features, such as noise in the proximity of the solution) is of order $\sim 1$, and that differences in the objective less than $10^{-3}$ are negligible. For this reason, BADS is *not* invariant to arbitrary rescalings of the objective $f$. This assumption does not limit the actual values taken by the objective across the optimization. If the objective $f$ is the log likelihood of a dataset and model (e.g., summed over trials), these assumptions are generally satisfied. They would not be if, for example, one were to feed to BADS the *average* log likelihood per trial, instead of the total (summed) log likelihood. In cases in which $f$ has an unusual scale, we recommend to rescale the objective such that the magnitude of differences of interest becomes of order $\sim 1$.

**Initialization** We initialize $\Delta_0^{\text{poll}} = 1$ and $\Delta_0^{\text{mesh}} = 2^{-10}$ (in standardized space). The initial design comprises of the provided starting point $\boldsymbol{x}_0$ and $n_{\text{init}} = D$ additional points chosen via a low-discrepancy Sobol quasirandom sequence [12] in the standardized box, and forced to be on the mesh grid. If the user does not specify whether $f$ is deterministic or stochastic, the algorithm assesses it by performing two consecutive evaluations at $\boldsymbol{x}_0$. For all practical purposes, a function is deemed noisy if the two evaluations at $\boldsymbol{x}_0$ differ more than $1.5 \cdot 10^{-11}$.[1]

## B.2 SEARCH stage

In BADS we perform an aggressive SEARCH stage in which, in practice, we keep evaluating candidate points until we fail for $n_{\text{search}}$ consecutive steps to find a *sufficient* improvement in function value, with $n_{\text{search}} = \max\{D, \lfloor 3 + D/2 \rfloor\}$; and only then we switch to the POLL stage. At any iteration $k$, we define an improvement *sufficient* if $f_{\text{prev}} - f_{\text{new}} \geq (\Delta_k^{\text{poll}})^{3/2}$, where $\Delta_k^{\text{poll}}$ is the poll size.

In each SEARCH step we choose the final candidate point to evaluate, $\boldsymbol{x}_{\text{search}}$, by performing a fast, approximate optimization of the chosen acquisition function in the neighborhood of the incumbent $\boldsymbol{x}_k$, using a two-step evolutionary heuristic inspired by CMA-ES [13]. This local search is governed by a *search covariance matrix* $\boldsymbol{\Sigma}$, and works as follows.

**Local search via two-step evolutionary strategy** We draw a first generation of candidates $\boldsymbol{s}_{\text{I}}^{(i)} \sim \mathcal{N}(\boldsymbol{x}_k, (\Delta_k^{\text{poll}})^2 \boldsymbol{\Sigma})$ for $1 \leq i \leq n_{\text{search}}$, where we project each point onto the closest mesh point (see Section 2.1 in the main text); $\boldsymbol{\Sigma}$ is a search covariance matrix with unit trace,[2] and $n_{\text{search}} = 2^{11}$ by default. For each candidate point, we assign a number of offsprings inversely proportionally to the square root of its ranking according to $a(\boldsymbol{s}_{\text{I}}^{(i)})$, for a total of $n_{\text{search}}$ offsprings [13]. We then draw a second generation $\boldsymbol{s}_{\text{II}}^{(i)} \sim \mathcal{N}(\boldsymbol{s}_{\text{I}}^{(\pi_i)}, \lambda^2 (\Delta_k^{\text{poll}})^2 \boldsymbol{\Sigma})$ and project it onto the mesh grid, where $\pi_i$ is the index of the parent of the $i$-th candidate in the 2nd generation, and $0 < \lambda \leq 1$ is a zooming factor (we choose $\lambda = 1/4$). Finally, we pick $\boldsymbol{x}_{\text{search}} = \arg\min_i a(\boldsymbol{s}_{\text{II}}^{(i)})$. At each step, we remove candidate points that violate non-bound constraints ($c(\boldsymbol{x}) > 0$), and we project candidate points that fall outside hard bounds to the closest mesh point inside the bounds.

**Hedge search** The search covariance matrix can be constructed in several ways. Across SEARCH steps we use both a diagonal matrix $\boldsymbol{\Sigma}_{\boldsymbol{\ell}}$ with diagonal $\left(\ell_1^2/|\boldsymbol{\ell}|^2, \ldots, \ell_D^2/|\boldsymbol{\ell}|^2\right)$, and a matrix $\boldsymbol{\Sigma}_{\text{WCM}}$ proportional to the weighted covariance matrix of points in $\mathbf{X}$ (each point weighted according to a function of its ranking in terms of objective values $y_i$, see [13]). At each step, we compute the probability of choosing $\boldsymbol{\Sigma}_s$, with $s \in \{\boldsymbol{\ell}, \text{WCM}\}$, according to a *hedging* strategy taken from the *Exp3* HEDGE algorithm [14],

$$p_s = \frac{e^{\beta_{\text{H}} g_s}}{\sum_{s'} e^{\beta_{\text{H}} g_{s'}}} (1 - \gamma_{\text{H}} n_{\boldsymbol{\Sigma}}) + \gamma_{\text{H}} \tag{S10}$$

where $\beta_{\text{H}} = 1$, $\gamma_{\text{H}} = 0.125$, $n_{\boldsymbol{\Sigma}} = 2$ is the number of considered search matrices, and $g_s$ is a running estimate of the reward for option $s$. The running estimate is updated each SEARCH step as

$$g_s^{\text{new}} = \alpha_{\text{H}} g_s^{\text{old}} + \frac{\Delta f_s}{p_s \Delta_k^{\text{poll}}} \tag{S11}$$

where $\alpha_{\text{H}} = 0.1^{1/(2D)}$ is a decay factor, and $\Delta f_s$ is the improvement in objective of the $s$-th strategy (0 if $s$ was not chosen in the current SEARCH step). This method allows us to switch between searching along coordinate axes ($\boldsymbol{\Sigma}_{\boldsymbol{\ell}}$), and following an approximation of the local curvature around the incumbent ($\boldsymbol{\Sigma}_{\text{WCM}}$), according to their track record of cumulative improvement.

## B.3 POLL stage

We perform the POLL stage only after a SEARCH stage that did not produce a *sufficient* improvement after $n_{\text{search}}$ steps. We incorporate the GP approximation in the POLL in two ways: when constructing the set of polling directions $\mathbf{D}_k$, and when choosing the polling order.

**Set of polling directions**   At the beginning of the POLL stage, we generate a preliminary set of directions $\mathbf{D}'_k$ according to the random LTMADS algorithm [15]. We then transform it to a *rescaled* set $\mathbf{D}_k$ based on the current GP kernel length scales: for $\boldsymbol{v}' \in \mathbf{D}'_k$, we define a rescaled vector $\boldsymbol{v}$ with $v_d \equiv v'_d \cdot \omega_d$, for $1 \le d \le D$, and $\omega_d \equiv \min\{\max\{10^{-6}, \Delta_k^{\mathrm{mesh}}, \ell_d/\mathrm{GM}(\boldsymbol{\ell})\}, \mathtt{UB}_d - \mathtt{LB}_d\}$, where $\mathrm{GM}(\cdot)$ denotes the geometric mean, and we use $\mathtt{PLB}_d$ (resp. $\mathtt{PUB}_d$) whenever $\mathtt{UB}_d$ (resp. $\mathtt{LB}_d$) is unbounded. This construction of $\mathbf{D}_k$ deviates from the standard MADS framework. However, since the applied rescaling is bounded, we could redefine the mesh parameters and the set of polling directions to accomodate our procedure (as long as we appropriately discretize $\mathbf{D}_k$). We remove from the poll set points that violate constraints, if present.

**Polling order**   Since the POLL is opportunistic, we evaluate points in the poll set starting from most promising, according to the ranking given by the chosen acquisition function [16].

## B.4   Update and termination

If the SEARCH stage was successful in finding a sufficient improvement, we skip the POLL, move the incumbent and start a new iteration, without changing the mesh size (note that mesh expansion under a success is not required in the MADS framework [15]). If the POLL stage was executed, we verify if overall the iteration was successful or not, update the incumbent in case of success, and double (halven, in case of failure) the mesh size ($\tau = 2$). If the optimization has been *stalling* (no sufficient improvement) for more than three iterations, we *accelerate* the mesh contraction by temporarily switching to $\tau = 4$.

The optimization stops when one of these conditions is met:

- the poll size $\Delta_k^{\mathrm{poll}}$ goes below a threshold $\Delta_{\mathrm{min}}^{\mathrm{poll}}$ (default $10^{-6}$);
- the maximum number of objective evaluations is reached (default $500 \times D$);
- the algorithm is *stalling*, that is there has no *sufficient* improvement of the objective $f$, for more than $4 + \lfloor D/2 \rfloor$ iterations.

The algorithm returns the optimum $\boldsymbol{x}_{\mathrm{end}}$ (transformed back to original coordinates) that has the lowest objective value $y_{\mathrm{end}}$. For a noisy objective, we return instead the stored point with the lowest quantile $q_\beta$ across iterations, with $\beta = 0.999$; see Section 3.4 in the main text. We also return the function value at the optimum, $y_{\mathrm{end}}$, or, for a noisy objective, our estimate thereof (see below, Section B.5). See the online documentation for more information about the returned outputs.

## B.5   Noisy objective

For noisy objectives, we change the behavior and default parameters of the algorithm to offset measurement uncertainty and allow for an accurate local approximation of $f$. First, we:

- double the minimum number of points added to the GP training set, $n_{\mathrm{min}} = 100$;
- increase the total number of points (within radius $\rho$) to at least 200, regardless of $D$;
- increase the initial design set size to $n_{\mathrm{init}} = 20$ points;
- double the number of allowed stalled iterations before stopping.

**Uncertainty handling**   The main difference with a deterministic objective is that, due to observation noise, we cannot simply use the output values $y_i$ as ground truth in the SEARCH and POLL stages. Instead, we adopt a *plugin* approach [17] and replace $y_i$ with the GP latent quantile function $q_\beta$ [18] (see Eq. 3 in the main text). Moreover, we modify the MADS procedure by keeping an *incumbent set* $\{\boldsymbol{x}_i\}_{i=1}^k$, where $\boldsymbol{x}_i$ is the incumbent at the end of the $i$-th iteration. At the end of each POLL stage, we re-evaluate $q_\beta$ for all elements of the incumbent set, in light of the new points added to the cache which might change the GP prediction. We select as current (active) incumbent the point with lowest $q_\beta(\boldsymbol{x}_i)$. During optimization, we set $\beta = 0.5$ (mean prediction only), which promotes exploration. For the last iteration, we instead use a conservative $\beta_{\mathrm{end}} = 0.999$ to select the optimum $\boldsymbol{x}_{\mathrm{end}}$ returned by the algorithm in a robust manner. For a noisy objective, instead of the noisy measurement $y_{\mathrm{end}}$, we return either our best GP prediction $\mu(\boldsymbol{x}_{\mathrm{end}})$ and its uncertainty $s(\boldsymbol{x}_{\mathrm{end}})$, or, more conservatively, an estimate of $\mathbb{E}[f(\boldsymbol{x}_{\mathrm{end}})]$ and its standard error, obtained by averaging $N_{\mathrm{final}}$ function evaluations

at $\boldsymbol{x}_{\text{end}}$ (default $N_{\text{final}} = 10$). The latter approach is a safer option to obtain an unbiased value of $\mathbb{E}[f(\boldsymbol{x}_{\text{end}})]$, since the GP approximation may occasionally fail or have substantial bias.

**Noise estimate** The user can optionally provide a noise estimate $\sigma_{\text{est}}$ which is used to set the mean of the hyperprior over the observation noise $\sigma$ (see Table S1). We recommend to set $\sigma_{\text{est}}$ to the standard deviation of the noisy objective in the proximity of a good solution. If the problem has tunable precision (e.g., number of samples for log likelihoods evaluated via Monte Carlo), we recommend to set it, compatibly with computational cost, such that the standard deviation of noisy evaluations in the neighborhood of a good solution is of order 1.

## C Benchmark

We tested the performance of BADS on a large set of artificial and real problems and compared it with that of many optimization methods with implementation available in MATLAB (R2015b, R2017a).[3] We include here details that expand on Section 4.1 of the main text.

### C.1 Algorithms

| Package | Algorithm | Source | Ref. | Noise | Global |
|---------|-----------|--------|------|-------|--------|
| `bads` | Bayesian Adaptive Direct Search | GitHub page [4] | This | ✓ | ≈ |
| `fminsearchbnd` | Nelder-Mead (`fminsearch`) w/ bounded domain | File Exchange[5] | [19] | ✗ | ✗ |
| `cmaes` | Covariance Matrix Adaptation Evolution Strategy | Author's website[6] | [13] | ✗ | ≈ |
| — (`active`) | CMA-ES with active covariance adaptation | — | [20] | ✗ | ≈ |
| — (`noise`) | CMA-ES with uncertainty handling | — | [21] | ✓ | ≈ |
| `mcs` | Multilevel Coordinate Search | Author's website[7] | [22] | ✗ | ✓ |
| `snobfit` | Stable Noisy Optimization by Branch and FIT | Author's website[8] | [23] | ✓ | ✓ |
| `global` | GLOBAL | Author's website[9] | [24] | ✗ | ✓ |
| `randsearch` | Random search | GitHub page[10] | [25] | ✗ | ✓ |
| `fmincon` | Interior point (`interior-point`, default) | Opt. Toolbox | [26] | ✗ | ✗ |
| — (`sqp`) | Sequential quadratic programming | — | [27] | ✗ | ✗ |
| — (`active-set`) | Active-set | — | [28] | ✗ | ✗ |
| `patternsearch` | Pattern search | Global Opt. Toolbox | [29] | ✗ | ✗ |
| `ga` | Genetic algorithms | Global Opt. Toolbox | [30] | ✗ | ≈ |
| `particleswarm` | Particle swarm | Global Opt. Toolbox | [31] | ✗ | ≈ |
| `simulannealbnd` | Simulated annealing w/ bounded domain | Global Opt. Toolbox | [32] | ✗ | ≈ |
| `bayesopt` | Vanilla Bayesian optimization | Stats. & ML Toolbox | [1] | ✓ | ✓ |

Table S3: **Tested algorithms.** *Top*: Freely available algorithms. *Bottom*: Algorithms in MATLAB's Optimization, Global Optimization, and Statistics and Machine Learning toolboxes. For all algorithms we note whether they explicitly deal with noisy objectives (*noise* column), and whether they are local or global algorithms (*global* column). Global methods (✓) potentially search the full space, whereas local algorithms (✗) can only find a local optimum, and need a multi-start strategy. We denote with (≈) *semi-local* algorithms with intermediate behavior – semi-local algorithms might be able to escape local minima, but still need a multi-start strategy.

The list of tested algorithms is reported in Table S3. For all methods, we used their default options unless stated otherwise. For BADS, CMA-ES, and `bayesopt`, we activated their *uncertainty handling* option when dealing with noisy problems (for CMA-ES, see [21]). For noisy problems of the CCN17 set, within the `fmincon` family, we only tested the best representative method (`active-set`), since we found that these methods perform comparably to random search on noisy problems (see Fig S1

right, and Fig 1, right panel, in the main text). For the combinatorial game-playing problem subset in the CCN17 test set, we used the settings of MCS provided by the authors as per the original study [33]. We note that we developed algorithmic details and internal settings of BADS by testing it on the CEC14 test set for expensive optimization [34] and on other model-fitting problems which differ from the test problems presented in this benchmark. For `bayesopt`, we allowed up to 300 training points for the GP, restarting the BO algorithm from scratch with a different initial design every 300 BO iterations (until the total budget of function evaluations was exhausted). The choice of 300 iterations already produced a large average algorithmic overhead of $\sim 8$ s per function evaluation. As acquisition function, we used the default EI-per-second [1], except for problems for which the computational cost is constant across all parameter space, for which we used the simple EI. All algorithms in Table S3 accept *hard* bound constraints `lb`, `ub`, which were provided with the BBOB09 set and with the original studies in the CCN17 set. For all studies in the CCN17 set we also asked the original authors to provide *plausible* lower/upper bounds `plb`, `pub` for each parameter, which we would use for all problems in the set (if not available, we used the hard bounds instead). For all algorithms, plausible bounds were used to generate starting points. We also used plausible bounds (or their range) as inputs for algorithms that allow the user to provide additional information to guide the search, e.g. the length scale of the covariance matrix in CMA-ES, the initialization box for MCS, and plausible bounds in BADS.

## C.2 Procedure

For all problems and algorithms, for the purpose of our benchmark, we first transformed the problem variables according to the mapping described in 'Transformation of variables and constraints' (Section B.1). In particular, this transformation maps the plausible region to the $[-1, 1]^D$ hypercube, and transforms to log space positive variables that span more than one order of magnitude. This way, all methods dealt with the same standardized domains. Starting points during each optimization run were drawn uniformly randomly from inside the box of provided plausible bounds.

For deterministic problems, during each optimization run we kept track of the best (lowest) function value $y_{\text{best}}^t$ found so far after $t$ function evaluations. We define the *immediate regret* (or error) at time $t$ as $y_{\text{best}}^t - y_{\text{min}}$, where $y_{\text{min}}$ is the true minimum or our best estimate thereof, and we use the error to judge whether the run is a success at step $t$ (error less than a given tolerance $\varepsilon$). For problems in the BBOB09 set (both noiseless and noisy variants), we know the ground truth $y_{\text{min}}$. For problems in the CCN17 set, we do not know $y_{\text{min}}$, and we *define* it as the minimum function value found across all optimization runs of all algorithms ($\approx 3.75 \cdot 10^5 \times D$ function evaluations per noiseless problem), with the rationale that it would be hard to beat this computational effort. We report the *effective* performance of an algorithm with non-negligible fractional overhead $o > 0$ by plotting at step $t \times o$ its performance at step $t$, which corresponds to a shift of the performance curve when $t$ is plotted in log scale (Fig 2 in the main text).[11]

For noisy problems, we care about the true function value(s) at the point(s) returned by the algorithm, since, due to noise, it is possible for an algorithm to visit a neighborhood of the solution during the course of the optimization but then return another point. For each noisy optimization run, we allowed each algorithm to return up to three solutions, obtained either from multiple sub-runs, or from additional outputs available from the algorithm, such as with MCS, or with population-based methods (CMA-ES, `ga`, and `particleswarm`). If more than three candidate solutions were available, we gave precedence to the main output of the algorithm, and then we took the two additional solutions with lowest observed function value. We limited the number of candidates per optimization run to allow for a fair comparison between methods, since some methods only return one point and others potentially hundreds (e.g., `ga`) – under the assumption that evaluating the true value of the log likelihood for a given candidate would be costly. For the combinatorial game-playing problem subset in the CCN17 set, we increased the number of allowed solutions per run to 10 to match the strategy used in the original study [33]. For noisy problems in the CCN17 set, we estimated the log likelihood at each provided candidate solution via 200 function evaluations, and took the final estimate with lowest average.

For plotting, we determined ranking of the algorithms in the legend proportionally to the overall performance (area under the curve), across iterations (deterministic problems) or across error tolerances (noisy problems.)

## C.3 Alternative benchmark parameters

In our benchmark, we made some relatively arbitrary choices to assess algorithmic performance, such as the range of tolerances $\varepsilon$ or the number of function evaluations. We show here that our findings are robust to variations in these parameters, by plotting results from the BBOB09 set with a few key changes (see Fig 1 in the main text for comparison). First, we restrict the error tolerance range for deterministic functions to $\epsilon \in [0.1, 1]$ instead of the wider range $\epsilon \in [0.01, 10]$ used in the main text (Fig S1 left and middle). This narrower range covers realistic practical requirements for model selection. Second, we reran the BBOB09 noisy benchmark, allowing $500 \times D$ functions evaluation, as opposed to $200 \times D$ in the main text (Fig S1 right). Our main conclusions do not change, in that BADS performs on par with or better than other algorithms.

Figure S1: **Artificial test functions (BBOB09).** Same as Fig 1 in the main text, but with with alternative benchmark parameters (in bold). *Left & middle*: Noiseless functions. Fraction of successful runs ($\varepsilon \in [\mathbf{0.1}, \mathbf{1}]$) vs. # function evaluations per # dimensions, for $D \in \{3, 6, 10, 15\}$ (96 test functions); for different BADS configurations (*left*) and all algorithms (*middle*). *Right*: Heteroskedastic noise. Fraction of successful runs at $\mathbf{500 \times D}$ objective evaluations vs. tolerance $\varepsilon$.

# D Numerical implementation

BADS is currently freely available as a MATLAB toolbox, `bads` (a Python version is planned).

The basic design of `bads` is simplicity and accessibility for the non-expert end user. First, we adopted an interface that resembles that of other common MATLAB optimizers, such as `fminsearch` or `fmincon`. Second, `bads` is *plug-and-play*, with no requirements for installation of additional toolboxes or compiling `C/C++` code via `mex` files, which usually requires specific expertise. Third, `bads` hides most of its complexity under the hood, providing the standard user with thoroughly tested default options that need no tweaking.

For the expert user or developer, `bads` has a modular design, such that POLL set generation, the SEARCH oracle, acquisition functions (separately for SEARCH and POLL), and initial design can be freely selected from a large list (under development), and new options are easy to add.

**GP implementation** We based our GP implementation in MATLAB on the GPML Toolbox [35] (v3.6), modified for increased efficiency of some algorithmic steps, such as computation of gradients,[12] and we added specific functionalities. We optimize the GP hyperparameters with `fmincon` in MATLAB (if the Optimization Toolbox is available), or otherwise via a the `minimize` function provided with the GPML package, modified to support bound constraints.

## Supplementary references

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

[29] Kolda, T. G., Lewis, R. M., & Torczon, V. (2003) Optimization by direct search: New perspectives on some classical and modern methods. *SIAM Review* **45**, 385–482.

[30] Goldberg, D. E. (1989) *Genetic Algorithms in Search, Optimization & Machine Learning*. (Addison-Wesley).

[31] Eberhart, R. & Kennedy, J. (1995) A new optimizer using particle swarm theory. *Proceedings of the Sixth International Symposium on Micro Machine and Human Science, 1995 (MHS'95).* pp. 39–43.

[32] Kirkpatrick, S., Gelatt, C. D., Vecchi, M. P., et al. (1983) Optimization by simulated annealing. *Science* **220**, 671–680.

[33] van Opheusden, B., Bnaya, Z., Galbiati, G., & Ma, W. J. (2016) Do people think like computers? *International Conference on Computers and Games* pp. 212–224.

[34] Liang, J., Qu, B., & Suganthan, P. (2013) Problem definitions and evaluation criteria for the CEC 2014 special session and competition on single objective real-parameter numerical optimization.

[35] Rasmussen, C. E. & Nickisch, H. (2010) Gaussian processes for machine learning (GPML) toolbox. *Journal of Machine Learning Research* **11**, 3011–3015.

## Footnotes

[1]Since this simple test might fail, users are encouraged to actively specify whether the function is noisy.

[2]Unit trace (sum of diagonal entries) for $\boldsymbol{\Sigma}$ implies that a draw $\sim \mathcal{N}(0, \boldsymbol{\Sigma})$ has unit expected squared length.

[3]MATLAB's `bayesopt` optimizer was tested on version R2017a, since it is not available for R2015b.

[4]https://github.com/lacerbi/bads

[5]https://www.mathworks.com/matlabcentral/fileexchange/8277-fminsearchbnd--fminsearchcon.

[6]https://www.lri.fr/~hansen/cmaes_inmatlab.html

[7]https://www.mat.univie.ac.at/~neum/software/mcs/

[8]http://www.mat.univie.ac.at/~neum/software/snobfit/

[9]http://www.inf.u-szeged.hu/~csendes/index_en.html

[10]https://github.com/lacerbi/neurobench/tree/master/matlab/algorithms

[11]We did not apply this correction when plotting the results of vanilla BO (`bayesopt`), since the algorithm's performance is already abysmal even without accounting for the substantial overhead.

[12]We note that version 4.0 of the GPML toolbox was released while BADS was in development. GPML v4.0 solved efficiency issues of previous versions, and might be supported in future versions of BADS.