[Reviews · NeurIPS 2017]

Reviewer 1



This work proposes a Bayesian optimisation framework implemented by combining search-based optimisation strategies to Gaussian Process-based local approximations of the target function. The study has in important practical focus, as the authors provide a detailed description of the proposed optimisations scheme along with a clear illustration of parameters and numerical schemes. The experimental results, both on synthetic and real data, show that the proposed framework provides state-of-art performances on a number of optimisation problems, either for noiseless and noisy problems. The work seems solid and comes with important implementation details. Although the novelty of the proposed methodology is somehow limited, the reproducibility effort is rather remarkable and increases the scientific merit of the study. It remains to evaluate the effectiveness of the proposed framework in the high-dimensional setting, as the optimisation of the proposed GP surrogate function may become very expensive and prone to local minima. However, in the present version the proposed framework could already find interesting applications in several domains.

Reviewer 2



This paper presents a new optimization methods that combines Bayesian optimization applied locally with concepts from MADS to provide nonlocal exploration. The main idea of the paper is to find an algorithm that is suitable for the range of functions that are slightly expensive, but not enough to require the sample efficiency of standard Bayesian optimization. The authors applied this method for maximum likelihood computations within the range of a ~1 second. A standard critique to Bayesian optimization methods is that they are very expensive due to the fact that they rely on a surrogate model, like a Gaussian process that has a O(n^3) cost. The method presented in this paper (BADS) also rely on a GP. This paper solves the issue by computing the GP only of a local region, limited to 50+10D points. The paper ignores all the work that has been done in Bayesian optimization with much more efficient surrogate models, like random forests [A], Parzen estimators [B] or treed GPs [7], where available software shows that the computational cost is comparable to the one from BADS. It is known that those methods have worse global performance than GP based BO for problems in R^n, but given that this method uses local approximation, I would assume that the performance per iteration is also lower that GP-BO. Furthermore, because the main objective of BO is sample efficiency, some of the problems presented here could be solved with 50-100 iterations. Thus, being even more efficient that BADS as they would not require extra steps. In fact, optimized BO software [C] has a computational cost similar to the one reported here for BADS for the first 100 iterations. Note that [C] already have the rank-one updates implemented as suggested in section 3.2. The way the results are presented leaves some open questions: - Is the error tolerance relative or absolute? Are the problems normalized? What does it mean an error larger than 10? Is it 10%? - Given the fact that at the end of the plot, there is still 1/3 of the functions are unsolved. Without seeing the actual behaviour of the optimizers for any function, it is impossible to say if they were close, stuck in a local optimum, or completely lost... - Why using different plots for different problems? Why not doing the noisy case versus evaluations? - How is defined the "...effective performance of BADS by accounting for the extra cost..." from Figure 2? - If BO is used as a reference and compared in the text in many parts, why it is not included in the experiments? If the authors think it should not naively included for the extra cost, they could also "account for the extra cost", like in Figure 2. There is also a large set of optimization algorithms, mainly in the evolutionary computation community, that relies on GPs and similar models for local or global modeling. For example: [D, E] and references therein. [A] Frank Hutter, Holger Hoos, and Kevin Leyton-Brown (2011). Sequential model-based optimization for general algorithm configuration, Learning and Intelligent Optimization [B] Bergstra, James S., Rémi Bardenet, Yoshua Bengio, and Balázs Kégl. "Algorithms for hyper-parameter optimization." In Advances in Neural Information Processing Systems, pp. 2546-2554. 2011. [C] Ruben Martinez-Cantin (2014) BayesOpt: A Bayesian Optimization Library for Nonlinear Optimization, Experimental Design and Bandits. Journal of Machine Learning Research, 15:3735-3739. [D] Zhou, Zongzhao, Yew Soon Ong, Prasanth B. Nair, Andy J. Keane, and Kai Yew Lum. "Combining global and local surrogate models to accelerate evolutionary optimization." IEEE Transactions on Systems, Man, and Cybernetics, Part C (Applications and Reviews) 37, no. 1 (2007): 66-76. [E] Jin, Yaochu. "Surrogate-assisted evolutionary computation: Recent advances and future challenges." Swarm and Evolutionary Computation 1, no. 2 (2011): 61-70.

Reviewer 3



In this paper the authors propose a Bayesian Optimization (BO) algorithm aimed at finding the optimum of a complex function (like non-differentiable, noisy, no closed form). The principle of the algorithm is relatively simple: approximate the objective function using a Gaussian Process and then move in a direction that would optimize an acquisition function (a combination of mean and variance at the potential next point). If this local search doesn’t improve the solution (little or no change in objective function value), the algorithm enters the poll stage that searches opportunistically in a larger region terminating as soon as a better solution is found. Function optimization is a problem of utmost importance and forms the backbone of many machine learning algorithms, so a novel method to approach the problem could be quite valuable. And even though some of the techniques used by the authors can be construed new applications, the overall contribution appears incremental. This assessment is partly based on the writing which is dense with too many ideas pushed into the supplementary material. I had a difficult time following all the details and at time just went ahead to avoid disrupting the reading flow. I also feel the authors could have utilized the space better and instead of spending almost two page describing previous studies and the parameter choices (pages 7 and 8), use the space to elaborate the ideas. Minor issue: should it be \Delta_k^{poll} in line 70? Given that the proposed method does outperform several benchmarks without incurring high cost and has potential to be further developed for use in case of high dimensionality, I am slightly inclined. However, I think the paper needs substantial amount of work to make it more readable.